# Mesenchymal Stem Cells for Spinal Cord Injury: Current Options, Limitations, and Future of Cell Therapy

**DOI:** 10.3390/ijms20112698

**Published:** 2019-05-31

**Authors:** Fabio Cofano, Marina Boido, Matteo Monticelli, Francesco Zenga, Alessandro Ducati, Alessandro Vercelli, Diego Garbossa

**Affiliations:** 1Department of Neuroscience “Rita Levi Montalcini”, Neurosurgery Unit, University of Turin, 10126 Turin, Italy; mmonticelli89@gmail.com (M.M.); zenga.francesco@gmail.com (F.Z.); aducati.nch@gmail.com (A.D.); dgarbossa@gmail.com (D.G.); 2Department of Neuroscience “Rita Levi Montalcini”, Neuroscience Institute “Cavalieri Ottolenghi”, University of Turin, Consorzio Istituto Nazionale di Neuroscienze, 10043 Orbassano, Italy; marina.boido@unito.it (M.B.); alessandro.vercelli@unito.it (A.V.)

**Keywords:** mesenchymal stem cells, spinal cord injury, regenerative medicine, translational medicine

## Abstract

Spinal cord injury (SCI) constitutes an inestimable public health issue. The most crucial phase in the pathophysiological process of SCI concerns the well-known secondary injury, which is the uncontrolled and destructive cascade occurring later with aberrant molecular signaling, inflammation, vascular changes, and secondary cellular dysfunctions. The use of mesenchymal stem cells (MSCs) represents one of the most important and promising tested strategies. Their appeal, among the other sources and types of stem cells, increased because of their ease of isolation/preservation and their properties. Nevertheless, encouraging promise from preclinical studies was followed by weak and conflicting results in clinical trials. In this review, the therapeutic role of MSCs is discussed, together with their properties, application, limitations, and future perspectives.

## 1. Introduction

Spinal cord injury (SCI) constitutes an inestimable public health issue, with an incidence of 40–80 per million people per year [1]. Generally, young adults are involved, where the burden of permanent neurological damage is unbearable for patients, for their caregivers, and for the health system. Prevention plays of course a key role, such as in cases of road accidents, criminal acts, or secondary causes (tumors, degenerative diseases); however, the real challenge involving scientists is about therapy, given the absence of a gold standard or effective treatment. Most of the post-traumatic degeneration of the nervous system is caused by multifactorial secondary damage including different molecular processes such as inflammation, neuronal death, ionic dysregulation, free radicals and lipid peroxidation, disconnection of normal nerve pathways, blood–brain barrier dysfunction, apoptosis, and necrosis, followed by cavitation processes and retrograde degeneration. In traumatic SCI, an early surgical decompression seems to be important in preventing secondary damage, in the range between 8 and 24 h after injury [2,3,4,5], together with spinal fixation to allow correct nursing and rehabilitation.

One of the most important and promising tested strategies involves the use of stem cells. Among them, mesenchymal stem cells (MSCs) are particularly appealing and showed hopeful promise in preclinical research, followed by weak and conflicting results in clinical trials. In this review, the therapeutic role of MSCs is discussed, together with their properties, application, limitations, and future perspectives.

## 2. Spinal Cord Injury

The spinal cord of mammals is organized in ten laminae of neurons, named dorsoventrally, according to the Rexed description (1952 and 1954) [6,7]. The neurons are mostly multipolar and vary in size. In the dorsal laminae, sensory neurons are found which receive inputs from the dorsal root ganglion cells and project to other spinal levels or to the upper centers of the sensory pathways. In the ventral laminae, cholinergic large motoneurons are devoted to the control of muscle contraction with motor axons. Somewhat in between, interneurons of different morphologies receive the descending projections and recurrent axonal fibers from spinal motoneurons, and influence motoneuron activity (Mai and Paxinos, 2011) [8]. Spinal cord neurons form intraspinal circuits which are controlled by descending pathways. The reflex arc is the most elementary intraspinal circuit.

The acute phase of SCI depends on the mechanism of trauma, which could be caused by contusion, laceration, stretch, compression, or direct massive destruction. The events related to the trauma constitute the primary injury, with disruption of neuronal pathways [9]. During the immediate phase (occurring within the first two hours) (Rowland et al., 2008) [10], neurons and glial cells at the lesion site die either by necrosis or by apoptosis (Zhang et al., 2012) [11]. Therefore, spinal cord repair should aim first to restore intraspinal circuits, and then to obtain regrowth of descending pathways to regain voluntary control of these intraspinal circuits.

It is well known that the most crucial phase in the pathophysiological process of SCI concerns the secondary injury, which is the uncontrolled and destructive cascade occurring later with aberrant molecular signaling, inflammation, vascular changes, and secondary cellular dysfunctions [12,13,14,15].

### 2.1. Secondary Injury

In the injured spinal cord, to a greater or lesser extent depending on the primary injury, a large amount of destructive processes upset the environment. Taking into account the vascular scenario, a global reduction of blood flow is observed, as a result of vasospasm, together with focal microhemorrhages or thrombosis, causing a global disfunction of the blood–spinal cord barrier [15,16]. The cascade of events also affects electrolytic homeostasis around cellular membranes and their ion pumps/transporters. Potassium (K^+^) increases its extracellular concentration, while sodium (Na^+^) and calcium (Ca^2+^) concentrations increase intracellularly [17,18]. This leads to the blockage of neuronal transmission. The influx of water caused by acidosis promotes cytotoxic edema followed by cellular death [18,19,20]. Many molecules are released, such as free radicals and neurotransmitters. The inflammatory process involves an immune response mediated by cellular invasion after disruption of the blood–spinal cord barrier. T cells, macrophages, microglia, and neutrophils infiltrate the neuronal tissue, acquiring a proinflammatory phenotype. The environment develops with the production of cytokines such as interleukin-1 beta (IL-1β), interleukin-1 alpha (IL-1α), tumor necrosis factor alpha (TNF-α), and interleukin-6 (IL-6), recruiting more cells in loco promoting neurodegeneration [21,22,23,24].

### 2.2. Chronic Phase and Neurodegeneration

The chronic phase is characterized by scar formation after gliosis and deposit of the extracellular matrix. Molecules with growth-inhibitory effects are released and target neuronal receptors. Oligodendrocyte death in the primary injury seems to be a crucial point in the SCI, because myelin debris contains inhibitory molecules preventing axonal growth in animal models, such as Nogo-A protein or myelin associated glycoprotein (MAG) [25,26,27,28]. Proteoglycans are also involved in the chronic phase and show a different pattern of functions in the pathophysiological process; while most of them constitute a limitation for axonal regrowth with their inhibitory features, others seem to border and limit the scar, preventing further amplification of tissue damage [29,30].

In this scenario, the removal of cellular debris and the cell environment is a key point for neuroregeneration; the modulation of macrophages, with their different phenotypes (M1 and M2) and effects in supporting neuroprotection or boosting inflammation, is then a multifactorial and crucial step in determining final outcomes [21,22,23,24].

Noteworthy, the regeneration of neurons within the injured spinal cord seems a pipe dream in mammals, but it is innate in the axolotl (salamander) where specific molecules may regulate glial reaction after SCI and promote the proliferation and migration of glial cells to replace the missing neural tube and stimulate axonal growth [31]. The identification of the cellular mechanisms which control neural regeneration is fundamental to promoting spinal cord repair after injury. The modulation of intraneuronal signaling networks and of the extracellular milieu is pivotal to enhance axonal regeneration, thus stimulating the regrowth of intraspinal circuits and of the descending and ascending pathways of the spinal cord [32]. The modulation of glial scar formation and of the alterations in the perineuronal nets, and the control of neuroinflammation following SCI are mandatory for spinal cord repair, even though far from being achieved [33]. Finally, axonal sprouting, synapse plasticity, and remodeling, in part cell-autonomous, may be differently regulated by many cells and molecules in the different compartments of the lesioned spinal cord [34].

## 3. Stem Cell Therapy and Appeal of MSCs

Stem cell division gives birth to an asymmetrical offspring with an additional progenitor cell and a daughter stem cell. A stem cell is able to differentiate into different phenotypes, thereby determining its potency. Totipotency is defined in the case where all terminal cell populations could be achieved, while multipotency describes the possibility to pursue a more restricted pattern of phenotypes.

The promotion of synapse formation or axon elongation by transplanted neuronal progenitors after damage was described in animal models [35]. Direct modulation in the differentiation of stem cells into terminal phenotypes expanded the focus of research, while promising studies showed encouraging recovery of neurological deficits after transplantation of derived cellular populations from embryonic stem cells (ESCs) in rodents after SCI [36,37,38,39,40].

Since then, researchers multiplied their fields of interest, ranging from the modulation of phenotypic pathways and optimization of transplant techniques, to imaging techniques in order to obtain spatial and temporal information on the grafts [41,42], up to clinical studies starting with the Geron clinical trial which promoted the use of human ESC-derived oligodendrocyte progenitor cells (OPCs) in the site of injury [43]. Although mechanisms are far from being elucidated, stem cell functions seem linked mostly to their paracrine effects and trophic support as shown in other neurological degenerative diseases [16,44,45,46]. Given by a relatively high number of studies focusing on SCI and neuronal repair, both in vivo and in vitro, evidence shows that a combinatory strategy involving not only stem cells, but also gene therapy, biomolecular targets and drugs, and biomaterials as scaffolds could dramatically improve the functional outcomes after SCI [16].

In this charming landscape, mesenchymal stem cells (MSCs) gained attention because of their easy isolation (from different sources) and preservation, raising no ethical concern [47,48,49], and of the limited risk of developing tumors [49]. In the case of ESCs, indeed many ethical controversies limited their application because of the problems related to the violation of a human embryo [50]. Additionally, MSCs maintain their regenerative potential even after cryopreservation at 80° C [51]. Their proliferation is very rapid, and a high multilineage differentiation can be obtained [47]. Immunoreactivity or a reaction versus hosts is minimal or absent.

Finally, MSCs show properties of “homing”, being able to migrate toward the site of lesion (Figure 1). According to other authors, we previously observed this phenomenon in SCI experimental models both by immunofluorescence reactions [52] and in MRI experiments [42]. Many authors demonstrated that this cellular behavior is mediated by several inflammatory or chemotactic factors [53]; for example, the vascular endothelial growth factor and hepatocyte growth factor, released at the injury level, can actually attract MSCs [53,54]. Additionally, the SDF-1*α*/CXCR4 (stromal cell-derived factor-1*α*/C–X–C chemokine receptor 4) axis plays an important role in these mechanisms [55]; the impairment or the upregulation of this axis can respectively affect or increase the MSC homing ability [56,57]. Other factors able to positively influence MSC migration include substance P [58] and the granulocyte colony-stimulating factor [59]. However, the precise mechanisms justifying the homing MSC ability are still largely unknown [54].

## 4. Secretome of MSCs

Although some differences were reported depending on the source, MSCs show a remarkable autocrine and paracrine activity [60,61] (Figure 2).

Through their secretome, MSCs can stimulate proliferation and differentiation of different cell types, including themselves. Notably, it was demonstrated that the release of growth factors, cytokines, and interleukins can also influence MSC migration (see also “homing” mechanism above), via an autocrine loop; indeed, when exposed to conditioned medium (i.e., the medium where MSCs are cultured), the MSC expression of Aquaporin 1 and CXCR4 (two membrane proteins involved in cell migration) significantly increased, by activating Akt and Erk intracellular signal pathways, and caused an enhancement of MSC migration [55].

Moreover, the MSC secretome can also exert immunomodulatory, anti-inflammatory, neurotrophic/neuroprotective and angiogenetic effects on the host microenvironment (as necessary in case of SCI).

The immunomodulation is realized thanks to the expression of the major histocompatibility complex-I on the MSC surface, in this way preventing T-cell recognition and induction of a host immune response [62]. Moreover, MSCs are able to inhibit the proliferation, the activation, and differentiation of T cells [63,64].

Concerning their anti-inflammatory potential, MSCs can secrete a variety of soluble molecules; among the anti-inflammatory cytokines, we can include tumor necrosis factor (TNF) β1, interleukin (IL)-13, IL-18 binding protein, ciliary neurotrophic factor (CNTF), neurotrophin 3 factor (NT-3), IL-10, IL-12p70, IL-17E, IL-27; moreover, MSCs can also modulate cytokine production of the host, for example, by inhibiting the release of pro-inflammatory cytokines (as interferon-γ and tumor necrosis factor α) or increasing the release of anti-inflammatory IL-10 [44,65].

To exert neuroprotection, MSCs secrete a number of neurotrophic factors, as brain-derived growth factor (BDNF), glial-derived growth factor (GDNF), nerve growth factor (NGF), NT-1, NT-3, CNTF, and basic fibroblast growth factor (bFGF) [44,65,66,67,68,69]; through these factors, MSCs can, on one side, prevent nerve degeneration and apoptosis, and, on the other, support neurogenesis, axonal growth, re-myelination, and cell metabolism [70,71,72,73,74,75,76].

MSCs can also induce angiogenesis, an important process by which new vasculature sprouts from pre-existing blood vessels; to this aim, MSCs secrete the tissue inhibitor of metalloproteinase-1, vascular endothelial growth factor, hepatocyte growth factor (HGF), platelet-derived growth factor (PDGF), IL-6, and IL-8. The production of these factors is particularly important for supporting the wound healing processes [77,78].

## 5. MSCs

MSCs can be obtained from different sources, each of which bears intrinsic characteristics differences, as shown below (Figure 2; Table 1) [52,79,80,81,82,83,84,85,86,87,88,89,90,91].

### 5.1. Bone Marrow Mesenchymal Stem Cells (BM-MSCs)

These cells are found within the adult bone marrow, where they contribute to hematopoiesis and bone regeneration. BM-MSCs can not only be obtained from humans, rodents, or primates, but also from several animal species such as sheep, dogs, cats, and bovines (Figure 3) [88,92,93,94,95,96,97,98].

The possibility to differentiate into cells of mesodermal origin and to adhere to plastic distinguishes BM-MSCs from hematopoietic cells. Their range of differentiation is larger than expected, including not only mesenchymal cells such as osteocytes, chondrocytes, and adipocytes, but also a broad range of lineages expressing non-mesenchymal markers [99,100]. Pre-clinical studies collected promising results (Table 2) Wislet-Gendebien et al. addressed the question of differentiation of MSCs in vitro trying to identify neuronal phenotypes. A series of markers were evaluated [100]. Authors found Nestin expression in some groups of cells, a marker for the responsive characteristic of MSCs to extrinsic signals. In Nestin-positive cells, they also registered an overexpression of proteins like sox2, sox10, pax6, fed, erbB2, and erb4. These cells showed a neuron-like conduction, responding to several neurotransmitters (GABA, glycine, glutamate). Compared to neurons, however, trains of action potentials or synaptic activities in co-cultured Nestin-positive MSCs were not observed.

BM-MSCs can not only be transplanted directly into the damaged spinal cord, but also infused with intravenous injections because of the aforementioned homing properties [101,102,103,104,105,106]. Deng et al. transplanted BM-MSCs two weeks after dorsal SCI in monkeys [104]. A partial functional improvement was noticed in terms of a slight motor recovery (active joints movements in the study group) and in electrophysiological studies with evoked potentials. Monkeys were then studied after three months with characterization of the scar. No neuronal cells were found. The analysis revealed the presence of markers such as neuron-specific enolase (NSE), the neurofilament (NF), and the glial fibrillary acidic protein (GFAP) in approximately 10% of the cells. The true blue, originally injected at the caudal side of injuries, was at the end traceable in the rostral thoracic spinal cord, red nucleus, and sensorymotor cortex. Zurita et al. transplanted BM-MSCs three months after dorsal SCI in pigs [105]. Authors used a motor score (0–10, with 10 considered as animals without deficits). Twelve weeks after transplantation, pigs that underwent stem cell therapy showed a mean score of 6.2 on the motor function scale. Some of the treated animals were even able to get up spontaneously. Recovery of evoked potentials was also noticed.

Many authors investigated BM-MSC transplantation in dogs [88,92,93,94,95]. Among them, Ryu et al. recorded improved neurological outcomes in MCSs groups after acute transplantation (one week after trauma), since all dogs had purposeful hind limb motion. He also showed that some MSCs expressed markers for neurons (NF160), neuronal nuclei (NeuN) and astrocytes (GFAP). NF160- and NeuN-positive neurons were found, and GFAP-positive reactive astrocytes were observed more often in the control group than in MSCs groups [88].

Hofstetter et al. in 2002 studied stem cell therapy after both iperacute and acute (one week) transplantation. Populations of neuron-like cells with the presence of neural markers were found, but they were not able to depolarize their membrane-like mature neurons. No clinical benefits were recorded. In the acute SCI group, a population of neuronal progenitors and astrocytes of the host were found in tissues after introduction of BM-MSCs in the lesion site [107].

Other preclinical studies are reported in Table 2 [108,109,110,111,112,113,114,115].

Jeon SR et al. described one of the first applications of these cells in patients with cervical SCI (Table 3). In this case, cells were isolated from iliac bones and then subjected to intramedullary or intradural introduction after expansion in a subacute and chronic state. After six months, most patients showed a slight improvement of motor function in the upper limbs, while magnetic resonance imaging (MRI) showed changes at the level of treatment in terms of the disappearance of the cavity margin and the presence of fiber-like streaks [116]. No evidence of neoplasm growth was observed even at three years follow-up [117] This study and others showed promising but very limited results (Table 3) [118,119,120,121,122,123,124,125,126]. Dai et al. [118] tested BM-MSCs in a randomized study with complete and chronic SCI patients. Neurological functions were evaluated with AIS grading, ASIA score, residual urine volume, and neurophysiological examination. In the treatment group (*N* = 20), 10 had clinical improvement. Mean motor improvement with AIS grading was 0.9 ± 1.07, that with the ASIA score was 11.5 ± 17.07, that with the sensory prick score was 5.2 ± 7.78, and that with the sensory light touch score was 5.4 ± 8.22. Residual urine volume (mL) was decreased with a mean of 61.55 ± 77.43. Patients were followed up for six months after an interval between the injury and stem cell therapy of 51.9 ± 18.3 months. No details about clinical improvements before stem cell therapy or other therapies were mentioned.

In a phase I/II controlled single-blind clinical trial, El-Kehir et al. [119] showed functional improvements over patients in the control group of stem cell therapy and physical therapy using AIS grading and ASIA scores in about half of the cases (46%), especially in patients with thoracic injuries with shorter durations of injury and smaller cord lesion. Motor recovery was recorded and promising but still qualitatively limited. Geffner et al. [120] described a partial efficacy of stem cell therapy with some improvements in ASIA, Barthel (quality of life), Frankel, and Ashworth scoring in eight cases (four acute, four chronic). Karamouzian et al. [121] described the results of a nonrandomized clinical trial of transplantation of BM-MSCs in 11 complete SCI patients against 20 in the control group. Results showed improvements of 45.5% of patients (a two-grade improvement from baseline, i.e., from ASIA A to ASIA C) in the study group vs. 15% in the control group, but were not statistically significant (*p* = 0.095). The heterogeneity and small number of the patients did not allow a reliable analysis. Mendonca et al. [122], Park et al. [123], and Cheng et al. (NCT01393977) described a slight functional improvement in small groups of patients treated with stem cells. Other papers and reviews questioned the extent of improvement and the correct timing of treatment [63,64]. Park et al. only described six cases with some improvements in motor function and changes in cord enhancement with the MRI [123]. Sykova et al. reported data from 20 patients with complete SCI who received transplants 10 to 467 days post-injury. Patients were then evaluated at three, six, and 12 months after implantation with ASIA protocol, the Frankel score, motor and somatosensory evoked potentials, and MRI evaluation of lesion size. Authors registered improvement in motor and/or sensory functions within three months in five of six patients with intra-arterial application, in five of seven acute patients, and in one of 13 chronic patients. Transplantation of cells appeared safe but there was no evidence that the observed beneficial effects were linked to cell therapy [124].

Therefore, new trials are needed, given the absence of protocols and the poor knowledge about mechanisms and outcomes of BM-MSC transplantation. In the ongoing trials (phase I and II), hundreds of patients should be enrolled, thus trying to improve quality of evidence.

Some studies are trying to explore benefits of different combinatory strategies involving not only BM-MSCs, but also technological tools such as virtual reality or exoskeletal stimulation to face the challenge with more holistic approaches [127].

BM-MSCs showed a very promising anti-inflammatory effect on cell environment. In animal models (rats), they promoted anti-inflammatory phenotypes of macrophages (M2) and suppressed lymphocyte proliferation before sustaining regeneration [128,129,130]. Furthermore, molecules such as vascular endothelial growth factor (VEGF), the glial cell-derived neurotrophic factor (GDNF), the nerve growth factor (NGF), and the brain-derived neurotrophic factor (BDNF) could be produced by MSCs and are currently related to the ability of MSCs to provide trophic support, studied in vivo with animal models [128,131,132]. This is probably why the genetically modified over-expression of these factors could improve clinical outcomes [133]. Finally, the homing properties of BM-MSCs could sustain targeted delivery of drugs acting like specific vectors [134].

In clinical trials involving SCI patients, BM-MSCs were injected with an intrathecal approach in about half of the cases, while, in the remaining studies, other routes of administration were used (in situ as grafts or with scaffold, intravenous or intramuscular) [16].

### 5.2. Umbilical Cord Mesenchymal Stem Cells (UC-MSCs)

These cells are obtained from cord blood or the umbilical cord [49], and grow in colonies with the support of growth factors [16,49]. The risk of graft rejection using these cells is very low as confirmed by studies demonstrating their hypoimmunogenicity [62]. In preclinical studies, using SCI animal models with rats or mouse, UC-MSCs showed a promising profile of neurotrophic, anti-apoptotic, and anti-inflammatory effects [135,136,137,138]. Molecular markers and neuron-like characteristics were observed after homogeneous maturation of UC-MSCs [139]. Despite the aforementioned results in pre-clinical studies, only few clinical trials were published describing minor improvements in some SCI patients [130,131,132]. Kang et al. described a case report of a young female patient with slight improvements after acute transplantation [140]. In the study of Yao et al., 25 patients with traumatic SCI (injury time >6 months) were treated with human umbilical cord blood stem cells via intravenous and intrathecal injection. The follow-up period was 12 months after transplantation. Results reported some autonomic restoration and changes in somatosensory evoked potentials [141]. The trial of Zhu et al. showed more promising, although limited results after transplantation in the chronic phase: 13 out of 20 patients improved their motor and sphincteric functions. Five out of 20 converted from complete to incomplete (two sensory, three motor; *p* = 0.038) SCI [142]. A phase II ongoing multicenter, randomized, sham-controlled trial (NCT03521336) recently started with patient enrollment, trying to evaluate efficacy of intrathecal transplantation of UC-MSCs. Completion of the study is expected in 2022.

### 5.3. Adipose-Derived Mesenchymal Stem Cells (AD-MSCs)

AD-MSCs represent an appealing source of transplantable MSCs, given the remarkable population of somatic stem cells and the availability of adipose tissue [143,144]. The ability of AD-MSCs to secrete growth factors, proteases, cytokines, extracellular matrix molecules, and immunomodulatory factors supports their potential of neuroregenerative, anti-apoptotic, angiogenetic, and wound healing actions [145]. Cellular survival pathways and repairing mechanisms in pre-clinical studies involved the upregulation of kinase proteins like ERK1/2 and Akt [40]. AD-MSC transplantation was studied in animal models showing no adverse effects but often unsatisfactory functional results [86,94,146]. Biomolecular and histological analysis revealed promising details. Kolar et al. [146] studied the effect of transplantation in rats with SCI. AD-MSCs were transplanted into the lateral funiculus 1 mm rostral and caudal to the C3–C4 lesion. In animals treated with cyclosporine, BDNF, vascular endothelial growth factor, and fibroblast growth factor-2 were expressed for about three weeks. An extensive ingrowth of 5HT-positive raphespinal axons was noticed in the trauma zone with some terminal arborizations reaching the caudal spinal cord. Furthermore, sprouting of raphespinal terminals in C2 contralateral ventral horn and C6 ventral horn on both sides was observed. Relative to the lesion scar, astrocytic processes extended into the middle of the trauma zone in association with regenerating axons. Menezes et al. described an abundant deposition of laminin at the lesion site and spinal midline, the appearance of cell clusters composed of neural-like precursors in the areas of laminin deposition, and the appearance of blood vessels [86]. Kim et al. showed a modification of the inflammatory environment after transplantation of AD-MSCs, with decreased astrogliosis-related signal molecules such as phosphorylated signal transducer and activator of transcription. Furthermore, markers like Tuj-1, Nestin, microtubule-associated protein 2, and neurofilament M were expressed as shown in other aforementioned studies [94]. In clinical studies, a slight sensory improvement was recorded in the majority of patients after intrathecal transplantation, but longitudinal clinical trials with concrete motor responses are still lacking [40]. Coadministration of other compounds, such as 17b-estradiol, and overexpression of Bcl-2 or chondroitinase ABC, were able to enhance therapeutic efficacy in dog models [147]. On this premis, a series of ongoing trials of AD-MSC transplantation is currently underway [40] (Table 3).

### 5.4. Amniotic Fetal Mesenchymal Stem Cells (AF-MSCs)

These cells can be obtained from amniotic membrane or amniotic fluid. Several features are attributed to AF-MSCs such as their multipotency, ease of isolation, and ability of proliferation, together with a low immunogenicity [148]. Despite this, only few pre-clinical studies in animal models were performed. They showed preliminary and limited results in terms of reduced inflammation and apoptosis, promoted angiogenesis, and provided trophic support [149,150,151]. No effective clinical trials followed pre-clinical investigations.

## 6. Biomaterials and Scaffolds for Stem Cell Therapy

Due to technological advances, researchers started to investigate biomaterials with the aim of promoting tissue repair, improving stem cell survival, and supporting their functions [152]. This strategy could be pursued using biomaterials as carriers, thereby ensuring stem cell biofactor delivery, or as a scaffold, offering a structural support for tissue regeneration [16].

Among synthetic polymers, biodegradable hydrogels (such as polylactic acid (PLA), polyglycol acid (PGA), and polyethylene glycol (PEG)) were developed to promote cellular survival and carry several advantages. They easily fill the lesion cavity after injection and show high flexibility, gas permeability, no toxicity, and a favorable mechanical profile [16]. Drugs, biomolecules, and biofactors can be loaded and released locally by hydrogels [153]. To improve proper micro-structure and ensure correct support, three-dimensional (3D) printing nano-architecture was developed to recreate a sustainable and attractive stem cell niche [16,154]. Usually, hydrogels are injected at the site of the lesion. The possibility to use them with a minimally invasive injection reduces the risk of a surgical procedure. Furthermore, they have the ability to load hydrophilic drugs and biomolecules with controlled release. Among the disadvantages, it is important to highlight that the kinetics and delivery of drugs could be inadequate because of uncontrolled diffusion or an unfavorable environment. For example, molecules with low steric hindrance cannot be controlled easily and might diffuse without reproducible control. Furthermore, the loading of hydrophobic drugs with a reduced affinity for the aqueous environment constitutes a real limitation. Bonds between drugs molecules could be built to increase or reduce the rate of release, depending on the ease of breakage of links, thereby offering controlled stem cell biofactor delivery [16].

Among natural scaffolds, a variety of materials were evaluated. Because of its biocompatibility, plasticity, and flexibility, fibrin was shown to promote regeneration and delay accumulation of astrocytes at the site of the injury. Enriched with stem cells or growth factors, fibrin improved survival and migration of transplanted cells, also increasing neural fiber density. Collagen and hyaluronic acid were proposed and used because of their elasticity, time of degradation, and ability to support cell adhesion and migration. Time of degradation plays a key role in letting the matrix produced by transplanted cells progressively replace the scaffold during the repairing process. An example of an innovative composite implant was described by Rochkind et al., cross-linked hyaluronic acid with growth factors and the adhesive molecule laminin (NVR-Gel), showing promising results. Natural scaffolds were able to reduce neuro-inflammation in the acute stage and support synaptic plasticity, as well as axonal outgrowth [106]. That said, despite this promising neuroengineering background, no concrete results were observed in clinical trials [16,155,156,157,158]. The field of polymeric scaffolds was developed to support stem cell survival and efficacy after transplantation, but favorable biomechanical properties did not translate encouraging pre-clinical studies into clinical success, highlighting the necessity for new comprehensive experimentations.

## 7. Limitations of Current Evidence and Future Directions

Stem cell technology is a growing and evolving field with an unquestionable appeal, as testified by many research papers and state-of-the-art reviews published in the English literature [16,40,80,158,159]. In our experience with MSCs or other stem cells [52,160], like in many other aforementioned papers, tests with animal models showed promising results. Despite this, many caveats arise and, thus, elicit caution against inordinate enthusiasm.

First of all, studies involving animal models are usually performed applying standardized protocols of lesions, treatments, and specific timings of transplantation in each group of investigation. These conditions are often inimitable in human patients with SCI, when timing and treatments are dependent on chance and emergency setting, or where lesions at the cord site could differ a lot from tailored laboratory damage. Most in vivo studies are necessarily performed with rodents, and, despite many anatomical or behavioral correspondences, human clinical trials should be the unavoidable aim of stem cell research.

Therefore, completed human trials showed only limited results. On the one hand, the use of MSCs in SCIs seems caused no harm; different trials [118,119,120,121,122,123,124,125,126,161] described, above all, the safety of stem cell therapy showing no adverse reactions or side effects. On the other hand, results in terms of clinical outcomes were poor compared to expectations. Among the others, few studies seemed particularly to encourage cell therapy.

Dai et al. [118] tested BM-MSCs in a randomized study with complete and chronic SCI patients. Neurological functions were evaluated with AIS grading, ASIA score, residual urine volume, and neurophysiological examination. In the treatment group (*N* = 20), 10 had clinical improvement. As already mentioned, patients were followed up for six months and details about clinical improvements before stem cell therapy or other therapies were not mentioned. Outcomes seemed limited, even if promising.

In the trial of El-Kehir et al. [119], functional improvements were noted but were confined and particularly involved patients with smaller and thoracic lesions. Geffner et al. [120] described partial efficacy of stem cell therapy with some improvements in ASIA, Barthel (quality of life), Frankel, and Ashworth scoring in eight cases (four acute, four chronic). The other clinical studies confirmed the trend of confined (Mendonca et al. [122], Park et al. [123], and Cheng et al. (NCT01393977)) or no significant improvements (Karamouzian et al. [121]). Many other trials failed to report satisfactory outcomes (NCT01186679, NCT02027246, NCT01769872, NCT01873547, NCT01624779, NCT01328860, NCT02237547, NCT01694927, and NCT01730183). There is a marked lack of large phase III trials of therapeutic efficacy, due to financial, ethical, and logistics reasons. [40] The phase III study of Oh et al. [162] showed weak efficacy in functional recovery, although some limitations could have compromised clinical results. For instance, only a single administration was given because of a restrictive government policy.

Finally, even if immunochemistry, molecular markers, and morphological tracts show that MSCs, once transplanted, present neuron-like characteristics, it is hard to consider them as such [16,129]; indeed, the expression of neuronal antigens can be simply due to the extremely immature nature of MSCs [163]. Moreover, cell fusion (between MSCs and neurons) was sometimes documented [164]; furthermore, when forced to transdifferentiate by chemical means (such as DMSO), MSCs showed evident morphological changes, which finally were simply attributed to cell shrinkage and changes in the cytoskeleton [165]. More sophisticated protocols are continuously being developed in order to differentiate MCSs into neurons [166,167]. However, currently, the efficacy of MSCs still seems in particular related to their paracrine activity, rather than to cellular replacement mechanisms [168].

Ongoing trials (Table 4) will probably help researchers improve knowledge about the clinical impact of stem cell therapy. Encouraging data from preclinical experiments were not concretely translated into clinical practice. This probably reflects the multifactorial and complex physiopathology of SCI, requiring a multimodal therapeutic approach. As a matter of fact, many points need to be further clarified and depicted, as listed below.
The optimal therapeutic protocols regarding the preparation, type, and number of stem cells transplanted;The timing of transplantation and route of administration;The paracrine effects and their influence on functional recovery;The importance of biomaterials and scaffold;The importance of microenvironment;The plasticity and ability to recreate connections of neuronal cells.Additionally, logistics, ethical, and financial problems related to this field of research constitute a real challenge to face in order to channel basic science studies into clinical practice.

## 8. Conclusions

MSC therapy represents an intriguing field of research trying to face the burden of SCI. MSCs of different origin, together with scaffolds, can release immunomodulating and neuroprotective factors which may support neuron survival, axonal growth, and control of glial scarring in absence of significant side effects. Despite promising preclinical findings, clinical trials failed to keep their promises and are still far from obtaining functional recovery and restoring neural circuits. Further studies are needed to improve our knowledge on their mechanisms of action and on the cellular mechanisms preventing restoration of neural circuits after SCI, while combinatory strategies involving stem cells, biomaterials, and modifications of cell environment could be the key to translate fascinating premises into clinical practice. A better relationship between preclinical and clinical studies with a back-and-forth approach is mandatory to enhance the efficacy of cell therapy. Nevertheless, stem cell therapy in SCI injury remains an experimental therapy, possibly in association with others, and should be tested and provided at no cost for the patient. Moreover, patients should be aware of the poor clinical results obtained thus far in clinical trials to prevent exaggerated expectations and dramatic psychological consequences in the case of failure to obtain significant results.

## Figures and Tables

**Figure 1 ijms-20-02698-f001:**
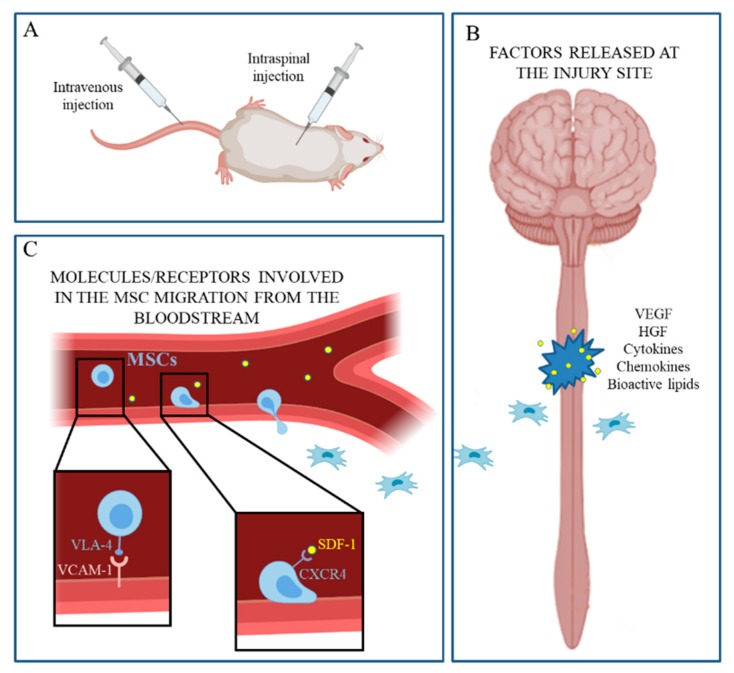
The main factors and mechanisms influencing mesenchymal stem cell (MSC) homing process are illustrated. (**A**) When injected either intravenously or intraspinally, MSCs show remarkable properties of “homing”. (**B**) At the injury site, some molecules (such as VEGF, HGF, cytokines, etc.) are secreted; when transplanted into the spinal parenchyma, MSCs are attracted by chemotactic stimuli and migrate toward the lesion site. (**C**) Moreover, when injected intravenously, MSCs can interact with endothelial cells through the VLA-4−VCAM-1 interaction; then, the extravasation is mediated by the interaction between the C–X–C chemokine receptor 4 and stromal cell-derived factor-1*α* (SDF-1), a chemotactic cytokine induced by proinflammatory stimuli. Created with BioRender software.

**Figure 2 ijms-20-02698-f002:**
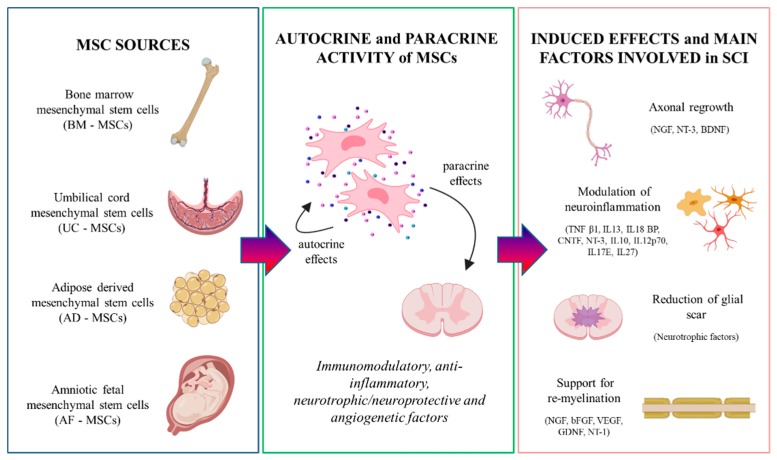
The main MSC sources, including bone marrow, umbilical cord, adipose tissue, and amnion. MSCs can exert both autocrine and paracrine effects. Among the molecules secreted, we can include several immunomodulatory and trophic factors, and anti-inflammatory cytokines; when transplanted in an injured spinal cord, the grafted cells can positively influence the host environment. Created with BioRender software.

**Figure 3 ijms-20-02698-f003:**
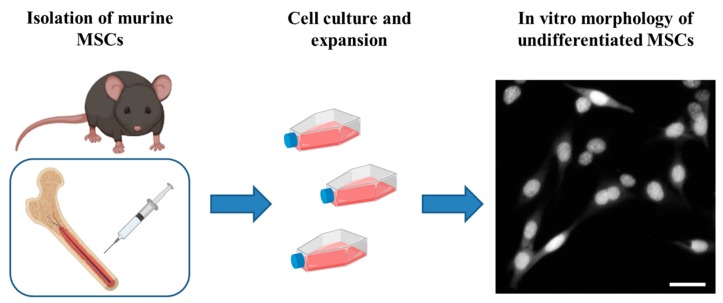
In preclinical experiments, bone marrow (BM)-MSCs can be isolated from femurs of adult mice and expanded in vitro; when cultured, the cells display the typical fibroblast-like shape. Created with BioRender software. Scale bar = 40 μm.

**Table 1 ijms-20-02698-t001:** Mesenchymal stem cell (MSC) characteristics.

MSC Type	Availability [83]	Invasive Procedure of Collection [83]	Cell Proliferation In Vitro [81,85]	Secretome * [79,82,87]	MSC Survival at the Injury Site After Graft [88,89]	Low Immunogenicity in the Host Tissue [84,85]	Anti-Inflammatory Effect in Injured Spinal Cord ** [84]	Glial Scar Reduction [52,80,86,90]	Axonal Regrowth/Sprouting Support [52,80,86,90,91]	Use in Pre-Clinical Studies (This Review)	Use in Clinical Trials (This Review)
BM-MSCs	+++	+++	++	+++	++	++	++	++	+++	+++	+++
UC-MSCs	+	not invasive	+++	+++	+++	+++	+++	++	+++	++	++
AD-MSCs	+++	++	++	+++	+++	++	++	++	+++	++	++
AF-MSCs	+	not invasive	+++	+++	+++	+++	not reported	not reported	+++	+	not reported

* Secretion of neurotrophic factors (bFGF, NGF, NT3, NT4, GDNF) is higher for UC-MSCs, whereas the production of pro-angiogenetic factors (VEGF, angiogenin, and PLGF) is higher for BM-MSCs and AD-MSCs. ** Based on the modulation of two inflammatory cytokines of the host tissue (COX-2 and IL-6).

**Table 2 ijms-20-02698-t002:** Main preclinical studies regarding MSCs.

Study	Type of Stem Cell Transplanted	Type of SCI	Animal	Administration	Scores	Adverse Reactions	Results	Cells Analysis/Findings in the Scar
Wislet-Gendebien et al. [100]	BM-MSC	In vitro study	N/A	N/A	Anti-glial fibrillary acidic protein (GFAP); anti-GLAST; anti-Tuj1; anti-NeuN; anti-SMI31; anti-MAP2ab; anti synaptophysi; anti-M2; anti-M6; RT-PCR, electrical conductivity	N/A	Neuron-like cells differentiated from Nestin + cells without the mature neuron electrical features; No differentiation in oligodendrocyte-like cells	Nestin + cells; GFAP + cells
Deng et al. [104]	BM-MSC	Transplantation of BM MSC 2 weeks after dorsal SCI	Macaco rhesus	Intralesional	Motor and sensitive improvement (Tarlov behavior assessment), SEP, MEP	None	Motor and sensitive functions improvement (Tarlov 2-3 achieved) in treated monkeys after 3 months follow up; Improvement of SEP and MEP	NSE +, NF +, GFAP + cells
Zurita et al. [105]	BM-MSC	BM MSC transplanted 3 months after dorsal SCI	Pigs	Intralesional	Clinical improvement (from 0 to 10 scale where 0 means paraplegia and 10 constantly useful hike), SEP, MRI	None	3 months after transplantation improvement of motor functions (mean score of 6.20) and SEP; reduction of the centromedullary cavity	GFAP +, NF +, S100 + cells
Hofstetter et al. [107]	BM-MSC	Iperacute (immediately after trauma) and acute (transplantation 1 week after dorsal trauma)	Lewis rat	Intralesional	Fibronectin, vimentin, laminin cells positivity; GFAP, electrical conduction	None	Markers of neuron-like cells, but no depolarization their membrane like mature neurons; No clinical benefit in the iper acute SCI group. In the acute SCI group Ab anti Nestin and GFAP of host astrocytes around and in the scar in the MSC treated population. Immature astrocytes Nestin + GFAP + with the possibility to differentiate into neuron-like cells	Neuron-like cells, host astrocytes closely connected with transplanted MSC cells, astrocyte-like cells
Nishio et al. [108]	HUCB stem cells	Acute (1 week after dorsal trauma)	Wistar rats	Intralesional	Basso, Beattie, Bresnahan locomotor scale (BBB), MRI	None	Hindlimb recovery, reduction of cystic cavity, no detection of any double-positive cells for human mitochondria and CD34, of CD4 positive cells, no significant differences between the two groups in the number of OX-42–positive or CD8-positive cells; GAP-43–positive fibers at the epicenter significantly higher than that of the control group	CD45 and human CD14, OX-42, CD4, CD8, GAP-43, 5-HT fibers, T-positive fibers
Pal et al. [109]	BM-MSC	Acute (1 week after dorsal trauma)	Wistar rats	Intrathecal	BBB locomotor scale, grid walk, plantar test, inclined plane; cells were tested for CD34, CD44, CD45, CD73, CD90, CD105 and HLA-DR.	None	Improved locomotor and sensory behavioral scores. Negative astroglial markers. No graft versus host immune reaction evoked by BM MSC, with the capacity to escape the immune system and be effective in wound healing	Negative astroglial markers, BM MSC
Nemati et al. [110]	Monkeys NSC	Acute (10 days after dorsal trauma)	Macaco rhesus monkeys	Intralesional	Spontaneous motor activity, Tarlov’s scale, limb pinch test, tail pinch test, sensory test, MRI, evaluation of neural specific markers Tuj1, MAP2, GFAP, Pax6, Sox1	None	Improvement in the sensory and motor activity, improvement in MRI	Isolated mNSCs express NSC markers such as nestin, Sox1, and Pax6 and could differentiate into mature neurons positive for MAP2 and GFAP
Gutierrez et al. [111]	Human fetal cortex-derived neural progenitor cells (hNPCs)	Iperacute (immediately after cervical trauma)	Göttingen minipig	Intralesional	Tarlov scale, sensory evaluation in the form of a tactile stimulus to the interdigital space	None	Improvement in motor and sensitive functions, no significant decrease in neuronal density between groups; cell engraftment ranged from 12% to 31%	
Hakim et al. [112]	BM-MSC	Acute (24 h after dorsal trauma)	Mice	Intralesional	Cells were evaluated by flow cytometry, immunohistochemistry, immunocytochemistry, proliferation assay differentiation assay, confocal microscopy and automatic cell quantification	None	MSCs transplanted downregulate genes related to cell-cycle and DNA metabolic/biosynthetic processes and upregulate genes related to immune system response, cytokine production, and phagocytosis/endocytosis; Sca1 and CD29, MHC I maintained expression; upregulated expression of CD45 and MHC II; Transplanted MSCs survived and proliferated to a low extent, no expression of Caspase-3, no differentiation into neurons or astrocytes	Transplanted MSCs express CD29, Sca1, and CD45 MHC-I and MHC-II; transplanted MSCs survive and proliferate but do not undergo apoptosis or neural differentiation
Cao et al. [113]	NSC	Acute (10 days after dorsal trauma)	Fischer rats	Intralesional, intrathecal	Cells were evaluated by immunohistochemistry, confocal microscopy and automatic cell quantification	None	The majority of transplanted cells either differentiated into GFAP + cells or remained nestin +. No Brd-U-positive neurons or oligodendrocytes detected	GFAP+ cells, nestin+ cells, Brd-U+ cells
Dasari et al. [114]	HUCB stem cells	Acute (1 week after dorsal trauma)	Lewis rat	Intralesional	BBB locomotor scale, cells were tested for CD44, NF200, CNPase, O1, beta III tubulin, APC, myelin basic protein caspase 3, MAP-2A&2B, confocal/fluorescence microscope, automatic cell quantification, immune blot	None	Improved locomotor and sensory behavioral scores, downregulation HUCB cellsmediated Fas and caspase	NF-200+ cells, CNPase+ cells, CD 44+ cells, co-localization of hUCB with neurons and oligodendrocytes
Cho et al. [115]	HUCB stem cells	Acute (1 week after dorsal trauma)	Sprague-Dawley rats	Intralesional	BBB locomotor scale, SSEPs, cells were evaluated by immunoistochemistry	None	Improved locomotor and sensory behavioral scores, shortened SSEPs latencies in treated rats	HuNu and GFAP + cells, MBP + cells, beta III tubular + cells
Khan et.al. [92]	AD-MSCs + BDNF	Acute (1 week after lumbar trauma)	Beagle dogs	Intralesional	BBB locomotor scale, cells were tested for Tuj-1, NF, GAP-43, GFAP, Nestin, COX2, TNFa, IL6, STAT3, IL-10, HO-1, BDNF	None	Significant improvement in hindlimb functions, with a higher BBB score	Increase in neuroregeneration, higher expression of Tuj-1, NF-M, and GAP-43, decreased expression of the inflammatory markers interleukin-6 (IL-6) and tumor necrosis factor-α (TNF-α), and an increased expression of interleukin-10 (IL-10). H&E staining showed more reduced intraparenchymal fibrosis
Ryu et. al. [88]	BM-MSC, AD-MSC, UCB MSC, Wharton’s jelly-derived MSC	Acute (1 week after lumbar trauma)	Beagle dogs	Intralesional	Olby score and Revised Modified Talov scale, BBB locomotor scale, confocal/fluorescence microscope. Immunoistochemistry	None	Significant differences of neurologic recovery in MSCs groups at 2 weeks following MSC transplantation. Purposeful hind limb motion of all dogs in the MSCs groups. No significant differences observed among the MSCs groups. UCB-derived MSCs (UCSCs) induced more nerve regeneration and anti-inflammation activity	Some MSCs expressed markers for neurons (NF160), neuronal nuclei (NeuN) and astrocytes (GFAP). NF160- and NeuN-positive neurons were found, GFAP-positive reactive astrocytes were observed more often in the control group than in MSCs groups. Lesion sizes were smaller, and fewer microglia and reactive astrocytes were found in the spinal cord epicenter of all MSC groups
Penha et.al. [93]	BM-MSC	Acute (10 days after dorsal or lumbar trauma)	Dogs	Intralesional	Clinical evaluation, MRI images	None	No changes at the MSC administration site into the spinal cord. Progressive recovery of the panniculus reflex and diminished superficial and deep pain response. Conscious reflex recovery occurred simultaneously with moderate improvement in intestine and urinary bladder functions	N/A
Kim et.al. [94]	AD-MSCs	Acute (1 week after dorsal or lumbar trauma)	Dogs	Intralesional	Clinical improvement: full recovery (normal neurologic state; grade 0), improved (regained deep pain perception (DPP) and recovery of ambulation, but still had mild ataxia; grade 1–2) and unsuccessful (did not regain DPP or the ability to walk without support; grade 3–5)	None	Clinical improvement (55.6% of the dogs were in full recovery, 22.2% showed improved outcomes and 22.2% had unsuccessful recovery)	N/A
Kim et.al. [95]	AD-MSCs	Iperacute (immediately after lumbar trauma)	Beagle dogs	Intravenous	Revised Tarlov scale, gait analysis, cells were evaluated by western blot	None	Significant enhanced motor function in AD-MSCs group compared with those in the control group at 7 days post treatment	The levels of GFAP, and GalCa were increased in the AD-MSC group, β3-tubulin levels were increased, COX-2, IL-6, and TNFα levels were significantly decreased; 3-NT level was significantly decreased, the level of 4-HNE was significantly decreased; the level of PC was significantly decreased

**Table 3 ijms-20-02698-t003:** Main clinical studies on BM-MSC transplantation.

Study	Type of SCI	Administration	n of Transplanted Cells	Transplanted Cells Type	Scores	Adverse Reactions	Results
Jeon et al. [116]	10 acute SCI patients	Intrathecal	8 × 10^6^ cells	Autologous MSCs	ASIA, Frankel score, EMG, SEP, MRI	None	Improvement in ASIA score, EMG, and SEP; improvement in MRI imaging
Dai et al. [118]	40 human patients; chronic and complete cervical SCI (AIS A)	Perilesional	Suspension with 8 × 10^5^ cells/microl	Autologous BM-MSCs expanded in culture	AIS, ASIA, residual urinary volume, EMG, MRI	None	45% AIS A to B; ASIA total scores were 31.6 prior and 43.1 after treatment (*p* = 0.01); preoperative urinary volume 235 mL to postop volume 173 mL (*p* = 0.01), improvement also in EMG and MRI
El Kheir et al. [119]	70 human patients; chronic complete cervical or thoracic SCI	Intrathecal	2 × 10^6^ cells/kg	Autologous BM-MSCs	AIS, ASIA, MRI, SEP	None	AIS conversion from AIS A to AIS B or C and from AIS B to AIC C; Improvement in ASIA score, SEP and in MRI. Higher improvement in the thoracic than in the cervical SCI group
Geffner 2008 [120]	8 human patients (4 acute SCI, 4 chronic SCI)	Directly into the spinal cord, directly into the spinal canal, and intravenous	/	Autologous BM-MSCs	ASIA, Barthel, Frankel Ashworth score, residual urinary volume, MRI	None	Improvement in all of the parameters
Karamouzian et al. [121]	11 human patients with acute or subacute (2-8 weeks after trauma) SCI	Intrathecal	7 × 10^5^ to 1.2 × 10^6^ cells	Autologous BM-MSCs	ASIA (12-33 months follow up)	None	Improvement in the ASIA score but the score was not statistically significant (*p* = 0.095)
Mendonca et al. [122]	14 human patients with chronic thoraco -lumbar SCI	Intralesional	5 × 10^6^ cells/cm^3^	BM-derived MSCs expanded in culture	ASIA, SEP, MRI, urodinamic, AIS	None	AIS A to B or C; incomplete injury; urinary function improved in 9 subjects, SEP improved in 1 subject
Park et al. [123]	6 human patients with cervical SCI treated at 72 h after trauma	Intralesional	2 × 10^8^ cells	Autologous BM-MSCs	Frankel, AIS, MRI	None	AIS A to B/C; improved MRI
Sykova et al. [124]	20 human patients with complete SCI transplanted from 10 to 467 days after trauma	Intra arterial vs. intra venous	89.7 +/− 70.7 × 10^6^ cells	Autologous BM-MSCs	Frankel, AIS, ASIA, SEP, MRI	None	Not significant results at 3–6–12 months follow-up; however, there was a positive trend
Pal et al. [125]	30 human patients with complete cervical or thoracic SCI	Intrathecal	1 × 10^6^ cells	Autologous BM-MSCs expanded in culture	ASIA, Barthel, SSEP, MEP, NCV, MRI	None	No significant results in ASIA score; variable patterns of recovery (especially in bladder functions), no significant variations in SSEP, MEP, NCV. Improved MRI
Moviglia et al. [126]	2 human patients with cervical and thoracic chronic SCI	Intra arterial	5 × 10^8^ to 1 × 10^9^ cells	Autologous BM-MSCs	SSEP, MEP, MRI, clinical examination	None	Improvement in all of the parameters

**Table 4 ijms-20-02698-t004:** Ongoing trials about MSCs. IANR-SCIRFS = International Association of Neural Restoration Spinal Cord Injury Functional Rating Scale; NSC = neural stem cell; SCIM III = Spinal Cord Independence Measure III; UC = umbilical cord.

ClinicalTrials.Gov Identifier	Title	MSC Type	Enrolled Subjects	Phase(s)	I End Point	II End Point	Date of Completion	Site of Administration	Intervention	Status
NCT03521336	Intrathecal transplantation of UC-MSC in patients with sub-acute spinal cord injury	UC-MSCs	130	II	ASIA score	IANR-SCIRFS score; EMG; residual urine	Dec 2022	Intrathecal	Allogeneic UC-MSCs	Recruiting
NCT03308565	Adipose stem cells for traumatic spinal cord injury	AD-MSCs	10	I	Acute adverse event	ASIA; MEPs; SSEPs; MRI; functional changes	Nov 2023	Intrathecal	Autologous AD-MSCs	Recruiting
NCT03225625	Stem cell spinal cord injury exoskeleton and virtual reality treatment study	BM-MSCs	40	N/A	ASIA score	ANS function; general well-being	Jul 2022	Paraspinal; intravenous; intranasal	Autologous BM-MSCs	Recruiting
NCT02917291	Safety and preliminary efficacy of FAB117-HC in patients with acute traumatic spinal cord injury	AD- MSCs	46	I/II	Safety	ISNC-SCI; SCIM III; SSEPs; MEPs	Jan 2020	Intramedullary	Autologous AD-MSCs	Recruiting
NCT01676441	Safety and efficacy of autologous mesenchymal stem cells in chronic spinal cord injury	BM-MSCs	32	II/III	Treatment	/	Dec 2020	Intramedullary	Autologous BM-MSC	Recruiting
NCT03505034	Intrathecal transplantation of UC-MSC in patients with late stage of chronic spinal cord injury	UC-MSCs	43	II	ASIA score	IANR-SCIRFS score; EMG; residual urine	Dec 2021	Intrathecal	Umbilical cord mesenchymal stem cells	Recruiting
NCT02574572	Autologous mesenchymal stem cell transplantation in cervical chronic and complete spinal cord injury	BM-MSCs	10	I	*N* of participants with treatment-related adverse events as assessed by MRI	ASIA score, ASIA impairment scale, improvement in sensorial mapping and neuropathic pain	Jun 2020	Intralesional	Autologous BM-MSC	Recruiting
NCT03521323	Intrathecal transplantation of UC-MSC in patients with early stage of chronic spinal cord injury	UC-MSCs	66	I/II	ASIA score	IANR-SCIRFS score; EMG; residual urine	Dec 2021	Intrathecal	Umbilical cord mesenchymal stem cells	Recruiting
NCT02574585	Autologous mesenchymal stem cell transplantation in thoracolumbar chronic and complete spinal cord injury	BM-MSCs	40	II	*N* of participants with treatment-related adverse events as assessed by MRI SCI	ASIA score, AIS score, improving in sensorial mapping and neuropathic pain	Jan 2022	Percutaneous	Autologous BM-MSC	Not yet recruiting

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
