# Peer review of "Mesenchymal Stem Cells for Spinal Cord Injury: Current Options, Limitations, and Future of Cell Therapy"

_ijms, 2019, doi:10.3390/ijms20112698_

Reviewer 1 Report

The manuscript attempts to summarize the current state of the art in the treatment of SCI using MSC, in preclinical and clinical studies. The text is well structured, but some improvements within the text are recommended to improve the significance of this review.

Page 2: The term “cellula” is not common and should be replaced by “cell”

Also other cell types than ESC and MSC are being used in SCI, and they should be shortly mentioned.

Fig. 1 This is not representative image of MSC. The cell population is heterogenous without typical fibroblastic shape. The typical MSC should be demonstrated instead of this not very good image with the shifted scale. 

Table 1: From what sources have made the authors these evaluations? E.g., why have UC MSC such a low availability in comparison to the others? What is the difference between low immunogenicity and survival? How was determined an anti-inflammatory effect? 

Fig. 2 The paracrine activity of MSC should be better described, with a basic overview of the secreted factors.

Table 2: this table shows just 4 studies. It should be improved with much broader overview of the published works. The other MSC sources (UC, AD) should be added to the table (or in an new table(s)) to overview preclinical state of the art.

Table 3: should be upgraded with more recent studies. The studies that used the application of bone marrow mononuclear cells are not differentiated from the studies that used expanded MSC. The number of cells should be added as well. The relation between the effect and the route of delivery should be discussed.

Table 4 shows all types of MSC in the ongoing clinical trials. The MSC types should be shown separately with more studies to give to the readers real overview.

Part 5: Biomaterials and Scaffolds…This part should be improved to better describe this field. Scaffold as carriers for cell delivery and their benefit should be more discussed.

E.g. the sentence “Usually hydrogels are injected intrathecally at the site of the lesion” is not true.

The sentence “Among the disadvantages, it is important to highlight that the kinetics and delivery of drugs could be inadequate because of uncontrolled diffusion or unfavorable environment.” is not clear.

In Part 6, the statement “Finally, even if immunochemistry, molecular markers and morphological tracts show that stem cells, once transplanted, present neuron-like features, they cannot be considered neurons.(11,76) The efficacy of stem cells do not seem then to be related with cellular replacement.” is not correct. Stem cells should be replaced by MSCs.

Also in part 7, the stem cells in the statement should be replaced by MSC.

Author Response

REVIEWER 1

1 - Page 2: The term “cellula” is not common and should be replaced by “cell”

Also other cell types than ESC and MSC are being used in SCI, and they should be shortly mentioned.

The wrong term cellula has been replaced, other types of stem cells are now mentioned as suggested. 

2 - Fig. 1 This is not representative image of MSC. The cell population is heterogenous without typical fibroblastic shape. The typical MSC should be demonstrated instead of this not very good image with the shifted scale. 

According to the referee, we replaced the figure 1 (NOTE that the previous Fig.1 becomes now Fig.2): the cell population is now homogenous and shows the typical fibroblast-like shape.

3 - Table 1: From what sources have made the authors these evaluations? E.g., why have UC MSC such a low availability in comparison to the others? What is the difference between low 

immunogenicity and survival? How was determined an anti-inflammatory effect? 

We are grateful to the referee for giving us the possibility to improve Table 1. To this aim, we added the lacking references. Based on these, we reported the data presented (as the availability of the different stem cell types).

With the term “immunogenicity” we refer to the possibility of activate the host immune system (UC-MSCs are less mmunogenic of others), whereas the term ”survival” refers to the MSC survival after graft. We have now better clarified these aspects in the Table 1.

The anti-inflammatory effect has been defined on the basis of the ability of MSCs to modulate host inflammatory molecules at the injury site, as reported by the literature.

We partially modified the Table 1 to make it clearer.

4 - Fig. 2 The paracrine activity of MSC should be better described, with a basic overview of the secreted factors.

Following to the referee suggestion, we better described the MSC paracrine activity by adding a new paragraph entitled “Secretome of MSCs”. We also modified the Figure 2 (NOTE that the previous Fig.2 becomes now Fig.1), citing the most relevant factors responsible for axonal regrowth, modulation of neuroinflammation, reduction of glial scar and re-myelination.

5 - Table 2: this table shows just 4 studies. It should be improved with much broader overview of the published works. The other MSC sources (UC, AD) should be added to the table (or in an new table(s)) to overview preclinical state of the art.

The table has been modified as suggested.

6 - Table 3: should be upgraded with more recent studies. The studies that used the application of bone marrow mononuclear cells are not differentiated from the studies that used expanded MSC. The number of cells should be added as well. The relation between the effect and the route of delivery should be discussed.

Studies has been differentiated according to the cells that have been used as suggested and their number. The discussion about the relation between the effect and the route of delivery is not possible since they are still ongoing studies. 

7 - Table 4 shows all types of MSC in the ongoing clinical trials. The MSC types should be shown separately with more studies to give to the readers real overview.

The table has been modified as suggested.

8 - Part 5: Biomaterials and Scaffolds…This part should be improved to better describe this field. Scaffold as carriers for cell delivery and their benefit should be more discussed.

The section has been improved as suggested with a more detailed description of biomaterials and scaffold

9 - E.g. the sentence “Usually hydrogels are injected intrathecally at the site of the lesion” is not true.

We modified this sentence.

10 - The sentence “Among the disadvantages, it is important to highlight that the kinetics and delivery of drugs could be inadequate because of uncontrolled diffusion or unfavorable environment.” is not clear.

The sentence has been modified and should be clear now. 

11 - In Part 6, the statement “Finally, even if immunochemistry, molecular markers and morphological tracts show that stem cells, once transplanted, present neuron-like features, they cannot be considered neurons.(11,76) The efficacy of stem cells do not seem then to be related with cellular replacement.” is not correct. Stem cells should be replaced by MSCs.

We better clarified this point and added some references. The term “stem cells” has been replaced by “MSCs”, as suggested.

12 - Also in part 7, the stem cells in the statement should be replaced by MSC.

Done as suggested

Reviewer 2 Report

The authors have reviewed the literature on the use of mesenchymal stem cells (MSCs) from different tissues for use in the treatment of spinal cord injury, specifically in relieving pain, neuron regeneration and the outcomes of clinical trials using MSCs. The review is well written and gives readers an overview of the current results in this area of research.

Further to the review being accepted for publication, this manuscript can be improved through answering the following questions,

The publication describes many outcomes of studies and the cell types. Can the authors include a paragraph/section on the cellular mechanisms controlling the regeneration of the neurons within the spinal cord ?

The authors mention the concept of cellular homing in their review and this is a potential repair mechanism in many of the described studies. Like in question 1, can the authors describe mechanisms for cellular homing or speculate on chemoattractants involved in cellular homing into the spinal tissue ? This has been mentioned in the manuscript but the authors should consider a separate section to discuss this concept.

The reviewer believes a paragraph summarizing the trophic factors produced in MSC should be included in the manuscript. Furthermore, this should also be included in Figure 2 to show a more detailed description on the trophic factors involved in the process of spinal tissue regeneration, as described in the manuscript. Thus, a paragraph or section with amended figure should be included in the revised manuscript.

The review focuses a lot on the neuronal regeneration within the spine. The authors should consider including a section on neurons within the spine and the structure of neurons in this region. A figure can also be included for this purpose

Author Response

REVIEWER 2

13 - The publication describes many outcomes of studies and the cell types. Can the authors include a paragraph/section on the cellular mechanisms controlling the regeneration of the neurons within the spinal cord ?

As requested, we commented this topic in the paragraph entitled “Chronic phase and neurodegeneration“.

14 - The authors mention the concept of cellular homing in their review and this is a potential repair mechanism in many of the described studies. Like in question 1, can the authors describe mechanisms for cellular homing or speculate on chemoattractants involved in cellular homing into the spinal tissue ? This has been mentioned in the manuscript but the authors should consider a separate section to discuss this concept.

We thank the referee for the suggestion: we deepened the description of the process of “cellular homing”, by adding some sentences at the end of the paragraph “Stem cells therapy and appeal of MSCs”.

15 - The reviewer believes a paragraph summarizing the trophic factors produced in MSC should be included in the manuscript. Furthermore, this should also be included in Figure 2 to show a more detailed description on the trophic factors involved in the process of spinal tissue regeneration, as described in the manuscript. Thus, a paragraph or section with amended figure should be included in the revised manuscript.

Following to the referee request, we inserted a new paragraph entitled “Secretome of MSCs”, where we mentioned the main immunomodulatory, anti-inflammatory, neurotrophic/neuroprotective and angiogenetic factors released by MSCs. Accordingly, we also modified the Figure 2, citing the most relevant molecules responsible for tissue regeneration. NOTE that the previous Fig.2 becomes now Fig.1.

16 - The review focuses a lot on the neuronal regeneration within the spine. The authors should consider including a section on neurons within the spine and the structure of neurons in this region. A figure can also be included for this purpose

According to the referee request, we included a description of the neurons of the spinal cord (motor neurons, sensory neurons, interneurons…) into the paragraph entitled “Spinal cord injury”.